# Rethinking Model Calibration Through Spectral Entropy Regularization in Medical Image Segmentation

**Kun Cheng**[1], **Yukun Zhang**[1], **William Henry Nailon**[2], **Tonggang Zhao**[1*]
[1]Beijing University of Posts and Telecommunications, Beijing, China
[2]NHS Lothian & The University of Edinburgh, Edinburgh, United Kingdom
kcheng@bupt.edu.cn, zyk@bupt.edu.cn, zhaotg@bupt.edu.cn
w.nailon@ed.ac.uk

## Abstract

Deep neural networks for medical image segmentation often produce overconfident predictions, posing clinical risks due to miscalibrated uncertainty estimates. In this work, we rethink model calibration from a frequency-domain perspective and identify two critical factors causing miscalibration: spectral bias, where models overemphasize low-frequency components, and confidence saturation, which suppresses overall power spectral density in confidence maps. To address these challenges, we propose a novel frequency-aware calibration framework integrating spectral entropy regularization and power spectral smoothing. The spectral entropy term promotes a balanced frequency spectrum and enhances overall spectral power, enabling better modeling of high-frequency boundary and low-frequency structural uncertainty. The smoothing module stabilizes frequency-wise statistics across training batches, reducing sample-specific fluctuations. Extensive experiments on six public medical imaging datasets and multiple segmentation architectures demonstrate that our approach consistently improves calibration metrics without sacrificing segmentation accuracy.

## 1 Introduction

Deep neural networks (DNNs) have become essential in medical image segmentation, enabling tasks such as tumor delineation and organ identification Hatamizadeh et al. (2021); Chen et al. (2021b); Isensee et al. (2021). Despite their high accuracy, DNNs often exhibit overconfident predictions—particularly near lesion boundaries, where predicted confidence substantially exceeds actual correctness Yeung et al. (2023); Larrazabal et al. (2023); Murugesan et al. (2025). This overconfidence undermines the reliability of automated systems in clinical settings and leads to misinformed diagnostic decisions Begoli et al. (2019).

To mitigate such risks, confidence calibration aims to align the predicted confidence with the true likelihood of correctness. Existing calibration methods in medical segmentation primarily operate in the spatial domain. Post-hoc techniques such as temperature scaling and Platt scaling apply dataset-level adjustments to logits Guo et al. (2017); Platt et al. (1999), but they often fail to adapt to anatomical and modality variability Tomani et al. (2021). Training-time methods, including label smoothing Müller et al. (2019), Focal Loss Mukhoti et al. (2020), and regional uncertainty regularization Murugesan et al. (2023a; 2024), add explicit or implicit constraints on output confidence Pereyra et al. (2017). While these approaches partially alleviate overconfidence, they predominantly focus on spatial domain characteristics and neglect potential frequency domain information, which can lead to suboptimal calibration performance.

Recent efforts suggest that frequency-domain information can enhance calibration and robustness. For instance, style-invariant Fourier domain adaptation has been used to stabilize model behavior across domains, and hybrid spatial-spectral representations have shown promise in improving

---

*Corresponding author

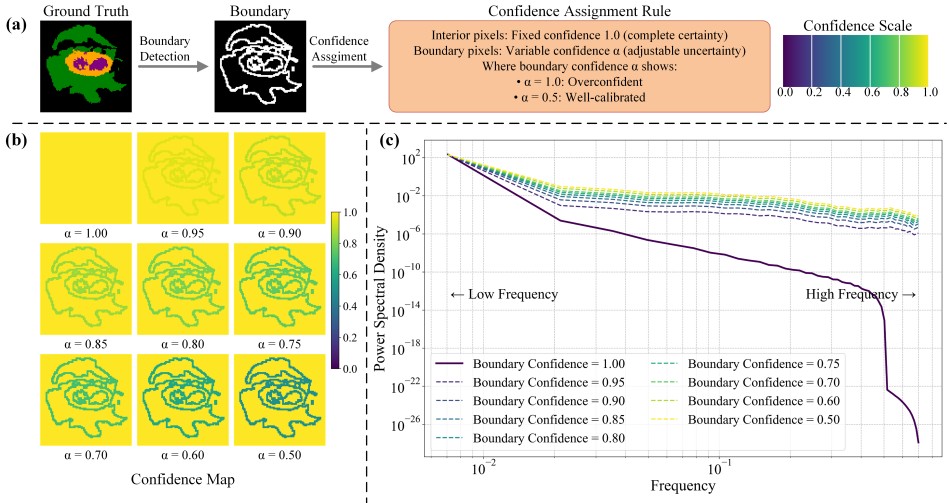

Figure 1: Frequency-domain analysis of overconfidence in medical image segmentation. **(a)** Synthesized confidence maps for a binary (foreground/background) segmentation example. Boundary pixels are assigned variable foreground confidence levels $\alpha$, while interior pixels have full confidence (confidence $= 1.0$). **(b)** Resulting confidence maps as boundary confidence $\alpha$ varies from overconfident (1.0) to well-calibrated (0.5). In a binary context, $\alpha = 0.5$ represents maximum uncertainty, the ideal state for a boundary pixel, as the model is equally confident in the foreground and background classes. Confidence values $\alpha < 0.5$ are not shown, as they would correspond to an incorrect background prediction, which is outside the scope of calibrating a correct foreground prediction. **(c)** Power spectral density (PSD) analysis. A Hamming window Hamming & Stearns (1979) is applied before the FFT to reduce spectral leakage caused by edge artifacts, ensuring a more accurate spectrum. As boundary confidence decreases toward the well-calibrated state, the PSD becomes richer across all frequency bands.

generalization and uncertainty estimation Chen et al. (2023b); Li et al. (2024); Tang et al. (2021). However, such works primarily target domain shift or style variation, rather than addressing overconfidence in confidence maps for medical segmentation.

From a spectral perspective, neural networks exhibit an inherent **Spectral Bias** during optimization, preferentially learning low-frequency patterns while inadequately capturing high-frequency details Rahaman et al. (2019); Xu et al. (2019), which further exacerbates overconfidence in boundary regions where high-frequency information is crucial for uncertainty estimation. Besides, as illustrated in Fig. 1, we create synthetic confidence maps by assigning different confidence levels to boundary pixels, while keeping interior pixels fixed. The power spectral density (PSD) analysis shows that overconfident maps (with boundary confidence $\alpha = 1.0$) have sparse spectral energy. In contrast, well-calibrated maps (e.g., $\alpha = 0.5$) display richer power across frequency bands. This suggests that overconfidence is linked to both spectral bias and low spectral complexity. Better-calibrated predictions distribute spectral power more evenly, capturing both high-frequency boundary uncertainty and low-frequency structural variations. We term this effect of spectral suppression in confidence maps as **Confidence Saturation**.

To address these two issues toward trustworthy medical image segmentation models, this paper introduces a novel adaptive calibration method from a spectral perspective. Our main contributions are as follows:

1. We conduct the first comprehensive spectral analysis of overconfidence in segmentation, revealing a strong link between prediction uncertainty and spectral power distributions.

2. We introduce a **power spectral smoothing** mechanism that stabilizes frequency-wise power statistics by aggregating spectral information across historical training batches, helping maintain segmentation performance while promoting consistent spectral patterns.

3. We design a **spectral entropy regularization** mechanism that enhances overall spectral power and improves frequency balance in confidence maps, leading to significantly improved calibration quality.

## 2 RELATED WORK

This section reviews research areas closely related to our approach, including confidence calibration methods, uncertainty estimation approaches, and frequency domain approaches in medical image analysis.

### 2.1 CONFIDENCE CALIBRATION METHODS

**Post-hoc Calibration Methods.** These methods improve calibration by adjusting output distributions after model training without retraining. Temperature Scaling Guo et al. (2017) adjusts softmax outputs by optimizing a global temperature parameter. Platt Scaling Platt et al. (1999) applies logistic regression to transform logits, while non-parametric methods such as Histogram Binning Zadrozny & Elkan (2001) and Isotonic Regression Fawcett & Niculescu-Mizil (2007) map predicted probabilities to empirical accuracies. However, these approaches lack region-specific calibration capabilities for different anatomical structures and show limited generalization under domain shifts across modalities Tomani et al. (2021). **In-training Calibration Methods.** These methods modify training objectives with regularization strategies to optimize calibration during training. Label Smoothing Müller et al. (2019) softens target labels to reduce overconfidence, while Focal Loss Mukhoti et al. (2020) reduces the influence of easily segmented samples. Entropy Regularization Pereyra et al. (2017) encourages higher predictive entropy to suppress overconfident predictions. Recent methods such as MarginLoss Murugesan et al. (2023a) and CRaC Murugesan et al. (2024) incorporate regional information and adaptive constraints for spatial-aware calibration. While these approaches demonstrate advantages in medical image tasks, they primarily rely on spatial domain features and lack systematic frequency domain analysis for calibration.

### 2.2 PROBABILISTIC UNCERTAINTY ESTIMATION METHODS

Uncertainty estimation approaches in deep learning are primarily based on probabilistic modeling. Bayesian neural networks (BNNs) Vadera et al. (2021), exemplified by MC-Dropout Gal & Ghahramani (2016), approximate Bayesian posteriors by retaining dropout during inference. Deep ensembles Lakshminarayanan et al. (2017) train multiple models with different initializations and aggregate predictions, while evidential deep learning Sensoy et al. (2018) directly parameterizes probability distributions rather than point estimates. These methods have been applied in medical image analysis, with Nair et al. (2020) evaluating approaches in brain tumor segmentation and Jungo et al. (2018) examining the impact of uncertainty estimation on segmentation accuracy. However, computational complexity and multiple inference requirements limit their real-time clinical applicability Murugesan et al. (2023b).

### 2.3 FREQUENCY DOMAIN APPROACHES IN MEDICAL IMAGE ANALYSIS

Frequency analysis has been widely applied in medical image analysis, from traditional Fourier-based denoising and enhancement to recent deep learning integration. Recent works have explored diverse applications including data augmentation impact analysis Yin et al. (2019), enhanced translation invariance Zhu et al. (2021), skin lesion segmentation optimization Li et al. (2021), MRI organ segmentation with spectral constraints Chen et al. (2023a), and cross-domain few-shot segmentation through frequency-aware matching Bo et al. (2025). Notably, Gorade et al. (2024) used a spectral loss that matches the prediction's spectrum to the ground truth's to improve segmentation accuracy. In addition, while Feng et al. (2023) and Chen et al. (2021a) also leverage frequency decomposition, they target domain adaptation and robustness via input/feature-level operations, whereas we directly regularize the spectrum of output confidence maps for calibration in a fully supervised segmentation setting. In contrast, we focus on confidence calibration and directly regularize the intrinsic spectral properties of the model's output itself, without relying on a ground truth spectrum. To our knowledge, this use of frequency analysis for confidence calibration in medical image segmentation remains underexplored.

# 3 METHOD

## 3.1 SPECTRAL ANALYSIS OF MODEL UNCERTAINTY

To motivate our frequency-aware calibration framework, we first conduct a comprehensive spectral analysis to investigate how model confidence relates to uncertainty estimation in medical image segmentation. Specifically, we analyze the power spectral density (PSD) of predicted confidence maps and observe that miscalibrated models often exhibit unbalanced spectral power distributions, characterized by the dominance of low-frequency components and suppressed high-frequency responses, as shown in Fig. 1. This **spectral bias** limits the model's ability to capture detailed boundary uncertainty and subtle anatomical structures. We observe that uncalibrated models often produce confidence maps that saturate toward uniformly high values, a behavior we define as **confidence saturation**. This effect results in reduced overall power spectral density (PSD), diminishing the frequency richness required to represent model uncertainty, especially in regions with boundary ambiguity. In contrast, well-calibrated models yield confidence maps with higher overall PSD, preserving richer spectral characteristics that better reflect structural variability and predictive uncertainty.

These findings reveal a strong connection between model calibration and the spectral characteristics of prediction confidence. The presence of spectral bias and confidence saturation suggests that current models may fail to adequately represent uncertainty across spatial scales, and particularly at high-frequency boundaries. This motivates the design of a calibration strategy that explicitly incorporates spectral structure, aiming to rebalance frequency distributions and increase overall spectral power, thereby enhancing model calibration and improving segmentation reliability.

## 3.2 SPECTRAL DECOMPOSITION AND WINDOW SMOOTHING

Building on our prior spectral uncertainty analysis, we now introduce a spectral decomposition and temporal smoothing scheme that extracts reliable frequency-domain confidence profiles, mitigating per-sample variability and enabling stable, frequency-aware calibration. Given input images $x \in \mathbb{R}^{B \times C_{\text{in}} \times D \times H \times W}$, where $B$ is the batch size, $C_{\text{in}}$ denotes the number of input channels, and $D, H, W$ denote depth, height, and width respectively, the neural network $f_\theta(\cdot)$ outputs prediction features (logits before softmax) $z = f_\theta(x) \in \mathbb{R}^{B \times C \times D \times H \times W}$, where $C$ is the number of classes. A scalar confidence map is constructed by taking the voxel-wise maximum logit value across the class dimension. While calibration is ultimately assessed on post-softmax probabilities, we operate on logits because they represent the network's raw, unnormalized evidence and offer a more sensitive signal for spectral analysis. Softmax outputs can saturate near 0 or 1, which can mask underlying spectral characteristics, whereas logits retain a wider dynamic range. This logit-based map therefore directly captures the precursors to overconfidence, where excessively high values are a primary cause of miscalibration.

$$z_b(d, h, w) = \max_{c=1,\ldots,C} z_{b,c}(d, h, w), \tag{1}$$

For each sample $b$, we transform $z_b(d, h, w)$ into the frequency domain via a 3D Fast Fourier Transform (FFT) to analyze its spectral properties:

$$F_b(u, v, w) = \text{SC}\left(\mathcal{F}_{3D}\left[z_b(d, h, w)\right]\right) \tag{2}$$

$$E_b(u, v, w) = \left|F_b(u, v, w)\right|^2, \tag{3}$$

where $(u, v, w)$ are frequency domain coordinates, and $E_b(u, v, w)$ represents the power spectral density at each frequency for sample $b$. Here, SC denotes Spectrum Centering, i.e., zero-frequency component shift to center the spectrum.

To characterize spectral behavior at different scales, we partition the frequency domain into $K$ concentric spherical shells $\mathcal{I}_k$ based on frequency magnitude, following the band decomposition strategy in Bo et al. (2025):

$$\mathcal{I}_k = \left\{(u, v, w) \;\middle|\; r_{k-1} < \sqrt{u^2 + v^2 + w^2} \leq r_k\right\}, \quad k = 1, \ldots, K, \tag{4}$$

with $r_k = \frac{k}{K} \frac{\min(D, H, W)}{2}$, which normalizes the frequency bands relative to the spatial dimensions, allowing scale-invariant spectral partitioning. This decomposition enables frequency-aware characterization of low-frequency structural content and high-frequency boundary information. For each

sample $b$ and frequency band $k$, we aggregate the spectral power within the corresponding shell:

$$S_b^{(k)} = \sum_{(u,v,w) \in \mathcal{I}_k} E_b(u,v,w). \tag{5}$$

To reduce statistical fluctuations and enhance training stability, we apply temporal smoothing over a sliding window of recent batches:

$$\widetilde{S}^{(k)} = \frac{1}{W} \sum_{w=1}^{W} \frac{1}{|\mathcal{B}_w|} \sum_{b \in \mathcal{B}_w} S_{b,w}^{(k)}, \tag{6}$$

where $W$ is the window size, $w = W$ corresponds to the current batch, and $w = 1$ corresponds to the oldest batch within the window. This smoothing aggregates spectral power statistics over multiple samples and batches to reduce noise and sample-specific variance. Let $\widetilde{S} = (\widetilde{S}^{(1)}, \ldots, \widetilde{S}^{(K)})$ denote the vector of smoothed spectral power across all $K$ frequency bands. During early training, when historical batches are unavailable, we replicate the current batch statistics to initialize the window and ensure numerical stability. This temporally smoothed spectral profile provides a consistent and robust reference for regulating spectral power distributions in frequency bands throughout training, thereby supporting frequency-aware calibration.

### 3.3 POWER SPECTRAL ENTROPY REGULARIZATION

Leveraging the temporally smoothed spectral profile vector $\widetilde{S}$, we introduce a power spectral entropy regularization term, which encourages the model to learn balanced frequency representations that alleviate spectral bias. It also promotes higher, more informative spectral power to counteract confidence saturation, thereby facilitating calibrated segmentation.

To quantify spectral diversity, we first convert the raw spectral power vectors into probability distributions. For the smoothed spectral profile $\widetilde{S}$ and each sample's profile $S_b$, we define their normalized counterparts, $\widetilde{P}$ and $P_b$, as:

$$P^{(k)} = \frac{S^{(k)}}{\sum_{j=1}^{K} S^{(j)} + \varepsilon}, \tag{7}$$

where $S$ is either $\widetilde{S}$ or $S_b$, and $\varepsilon$ is a small constant (e.g., $10^{-6}$) to ensure numerical stability in both the normalization and entropy calculation steps. The spectral entropy is then computed using the standard Shannon entropy formula:

$$H_{\text{spec}}(P) = -\sum_{k=1}^{K} P^{(k)} \log(P^{(k)} + \varepsilon). \tag{8}$$

Here, $P$ can be $\widetilde{P}$ or $P_b$. We select Shannon entropy as it is a principled measure from information theory for quantifying the uniformity of a probability distribution. A higher spectral entropy indicates that power is more evenly distributed across all frequency bands rather than being concentrated in a few. This directly aligns with our goal of encouraging richer, more balanced frequency representations for better uncertainty estimation. The entropy $H_{\text{spec}}(P_b)$ is computed independently for each sample $b$ in a batch. While this compresses the spectrum into a summary statistic, it intentionally quantifies the spectral uniformity of each sample to address spectral bias. By penalizing this per-sample value, our method enhances the frequency representation of each specific prediction, thereby preserving its unique characteristics rather than obscuring them.

To address spectral bias and the resulting confidence saturation, we introduce a hinge-like spectral regularization loss. It encourages each sample's spectral entropy $H_{\text{spec}}(P_b)$ to approach or exceed the dynamically updated target entropy $H_{\text{spec}}(\widetilde{P})$:

$$\mathcal{L}_{\text{Spectral}} = \frac{1}{B} \sum_{b \in B} \left[ \max\left(0, H_{\text{spec}}(\widetilde{P}) - H_{\text{spec}}(P_b)\right) \right]^2. \tag{9}$$

The mechanism of this loss directly counteracts spectral bias. The hinge formulation creates a targeted learning signal: a penalty is incurred only when a sample's spectral entropy is deficient

compared to the dynamic target. Minimizing this penalty formally encourages a more uniform spectral distribution by pushing power from dominant low-frequency bands into underrepresented high-frequency bands. While entropy itself is scale-invariant, this re-balancing forces the model to increase the total spectral energy of the logit map to avoid compromising the task loss. This prevents the generation of overly smooth, low-energy confidence maps, directly mitigating the effects of confidence saturation.

The $\max(\cdot, 0)$ operation ensures that only samples with spectral entropy below the adaptive target incur penalties. This selective penalization encourages the model to enhance frequency-domain diversity while maintaining calibration performance for already well-calibrated samples.

### 3.4 OVERALL TRAINING OBJECTIVE

We integrate our spectral regularization with the conventional segmentation objective through a weighted combination:

$$\mathcal{L}_{\text{total}} = \mathcal{L}_{\text{CE}} + \lambda \mathcal{L}_{\text{Spectral}}, \tag{10}$$

where $\mathcal{L}_{\text{CE}}$ is the cross-entropy loss for pixel-wise accuracy, and $\lambda > 0$ balances the trade-off between segmentation performance and uncertainty calibration. The hyperparameter $\lambda$ allows users to balance segmentation accuracy and calibration strength. This flexibility is important because different organs or lesions exhibit varying levels of uncertainty, requiring tailored calibration strategies to reflect the underlying uncertainty characteristics in each case optimally.

## 4 EXPERIMENTS

### 4.1 DATASETS AND IMPLEMENTATION DETAILS

We conducted experiments on six public medical imaging datasets, covering diverse anatomical structures and imaging modalities, to comprehensively evaluate our method. These datasets are Brain Tumor Segmentation (BraTS) 2020 Menze et al. (2014), Automated Cardiac Diagnosis Challenge (ACDC) Bernard et al. (2018), Fast and Low GPU memory Abdominal oRgan sEgmentation (FLARE) 2021 Ma et al. (2022), MICCAI Grand Challenge on 6-month Infant Brain MRI Segmentation (iSeg) 2017 Sun et al. (2021), A Tumour and Liver Automatic Segmentation (ATLAS) 2023 Quinton et al. (2023), and Prostate MR Image Segmentation (PROMISE) 2012 Litjens et al. (2014). All datasets were split into training, validation, and test sets with a ratio of 4:1:1.

Experiments were conducted on Ubuntu 22.04 with an NVIDIA RTX 4090 GPU, utilizing the U-Net architecture Ronneberger et al. (2015). We adopted consistent training settings across all experiments, including a batch size of 2, input patch size of $96 \times 96 \times 96$, and the SGD optimizer (initial learning rate 0.01). Input data were normalized to $[0, 1]$.

### 4.2 EVALUATION METRICS AND COMPARISON METHODS

Segmentation performance was evaluated using Dice Similarity Coefficient (DSC), 95% Hausdorff Distance (HD95), and Average Surface Distance (ASD), which collectively assess overlap accuracy, boundary error, and surface deviation, respectively. To evaluate calibration, we used Expected Calibration Error (ECE) as the primary metric, following the protocol in Nixon et al. (2019), and included Static Calibration Error (SCE) and Thresholded Adaptive Calibration Error (TACE). SCE generalizes ECE to multi-class settings via class-wise binning, while TACE improves robustness through equal-sized binning and probability thresholding. Both ECE and SCE were computed using $B = 15$ bins, and TACE employed a threshold $\epsilon = 0.001$ as in Murugesan et al. (2024).

All comparison methods are in-training calibration strategies built upon the cross-entropy loss, consistent with our approach. We implement these baselines using the optimal hyperparameter configurations reported in their respective original publications, a standard practice for fair benchmarking. These include Focal Loss (FL, $\gamma = 3$) Lin et al. (2017), Label Smoothing (LS, $\alpha = 0.1$) Szegedy et al. (2016), Margin-based Label Smoothing (MbLS, $\lambda = 0.1$, $m = 5$) Murugesan et al. (2023b), Spatially Variant Label Smoothing (SVLS, $\sigma = 2$) Islam & Glocker (2021), and Class and Region-Adaptive Constraints (CRaC, $\gamma = 1.2$, $\mu = 0.9$, $R = 2$, $margin = 5$) Murugesan et al. (2024).

# 5 RESULTS AND DISCUSSION

Table 1: Quantitative comparison of segmentation performance across multiple datasets (DSC↑, HD95↓, ASD↓). ↑ indicates higher is better, ↓ indicates lower is better. **Bold** denotes the best result, and *italics* indicate the second-best.

| Dataset | Method & Publication | | | | | | | | | | | | | | | | | | | | |
|---|---|---|---|---|---|---|---|---|---|---|---|---|---|---|---|---|---|---|---|---|---|
| | DSC↑ (%) | | | | | | | HD95↓ | | | | | | | ASD↓ | | | | | | |
| | CE | FL | LS | MbLS | SVLS | CRaC | Ours | CE | FL | LS | MbLS | SVLS | CRaC | Ours | CE | FL | LS | MbLS | SVLS | CRaC | Ours |
| | - | ICCV | CVPR | MedIA | IPMI | MICCAI | - | - | ICCV | CVPR | MedIA | IPMI | MICCAI | - | - | ICCV | CVPR | MedIA | IPMI | MICCAI | - |
| BraTS2020 | *86.9* | 84.6 | 86.7 | 85.9 | 85.9 | 85.9 | **87.2** | *2.27* | 2.80 | 2.52 | 2.91 | 2.46 | 2.58 | **2.23** | 0.40 | 0.44 | *0.37* | 0.41 | 0.47 | 0.39 | **0.37** |
| iSeg2017 | *94.2* | 94.0 | 94.1 | 94.1 | *94.2* | *94.2* | **94.4** | *1.01* | 1.46 | 1.13 | 1.15 | 1.15 | 1.17 | **0.85** | 0.06 | 0.06 | 0.06 | *0.06* | 0.06 | 0.06 | **0.05** |
| FLARE2021 | 91.5 | 87.7 | 91.3 | *91.5* | 91.2 | 91.4 | **92.5** | 8.29 | *6.17* | 8.37 | 7.38 | 8.40 | 7.48 | **4.55** | 2.64 | 2.22 | 2.49 | *2.09* | 2.22 | 2.10 | **1.34** |
| ACDC | *91.1* | 89.4 | 91.0 | 90.7 | *91.1* | 91.0 | **91.3** | *0.75* | 0.90 | 0.88 | 0.83 | 0.80 | 0.81 | **0.74** | 0.10 | 0.12 | 0.11 | *0.10* | 0.11 | 0.11 | **0.10** |
| ATLAS2023 | 68.7 | 65.9 | 69.1 | *70.5* | 68.3 | 69.8 | **71.8** | 22.20 | 20.19 | 19.76 | *18.84* | 20.18 | 20.31 | **17.82** | 4.99 | 4.53 | 4.46 | **3.54** | 5.21 | 3.87 | *3.57* |
| PROMISE2012 | *80.2* | 78.9 | 79.6 | 79.1 | 79.5 | 80.1 | **81.2** | 19.80 | *18.79* | 19.93 | 22.05 | 20.33 | 18.95 | **18.24** | 7.36 | 8.51 | 5.22 | 8.99 | 8.86 | *4.32* | **4.18** |

Table 2: Quantitative comparison of calibration performance across multiple datasets (ECE↓, SCE↓, TACE↓).

| Dataset | Method & Publication | | | | | | | | | | | | | | | | | | | | |
|---|---|---|---|---|---|---|---|---|---|---|---|---|---|---|---|---|---|---|---|---|---|
| | ECE↓ ($\times 10^{-3}$) | | | | | | | SCE↓ ($\times 10^{-3}$) | | | | | | | TACE↓ ($\times 10^{-3}$) | | | | | | |
| | CE | FL | LS | MbLS | SVLS | CRaC | Ours | CE | FL | LS | MbLS | SVLS | CRaC | Ours | CE | FL | LS | MbLS | SVLS | CRaC | Ours |
| | - | ICCV | CVPR | MedIA | IPMI | MICCAI | - | - | ICCV | CVPR | MedIA | IPMI | MICCAI | - | - | ICCV | CVPR | MedIA | IPMI | MICCAI | - |
| BraTS2020 | 9.1 | 5.5 | 25.2 | *1.9* | 2.1 | 2.2 | **1.5** | 10.7 | 6.4 | 13.8 | *4.0* | 5.0 | 5.1 | **0.8** | 42.7 | 37.7 | 61.4 | *31.5* | 32.8 | 39.4 | **20.2** |
| iSeg2017 | 4.5 | 6.5 | 17.7 | *2.1* | 2.6 | 2.6 | **2.0** | 11.9 | 4.0 | 8.5 | *2.3* | 2.9 | 2.9 | **2.0** | 15.6 | 17.6 | 33.8 | *11.8* | 13.5 | 13.4 | **6.4** |
| FLARE2021 | 25.5 | 19.4 | 6.5 | 2.2 | 2.5 | 2.4 | **0.8** | 18.1 | 17.3 | 13.0 | 11.5 | 9.1 | 9.4 | **0.6** | 37.8 | 52.7 | 80.0 | 30.4 | *24.2* | 28.5 | **3.1** |
| ACDC | 32.5 | 32.3 | 71.9 | *23.2* | 27.9 | 28.1 | **2.1** | 16.6 | 16.7 | 38.1 | *10.6* | 13.3 | 1.1 | **1.1** | 32.0 | 38.0 | 48.1 | 26.7 | 33.9 | **5.3** | *18.5* |
| ATLAS2023 | 24.9 | 9.4 | 50.4 | 7.2 | *6.8* | 7.0 | **5.5** | 15.2 | 22.9 | 44.3 | 9.6 | 22.7 | *5.5* | **4.2** | *18.1* | 40.7 | 89.9 | 20.8 | 37.8 | 37.0 | **15.9** |
| PROMISE2012 | *11.7* | 12.6 | 20.2 | 12.8 | 20.7 | 20.8 | **10.8** | 25.7 | 34.2 | 52.8 | *18.0* | 54.7 | 52.6 | **14.8** | 48.1 | 45.6 | 55.0 | *41.7* | 64.1 | **41.2** | 55.7 |

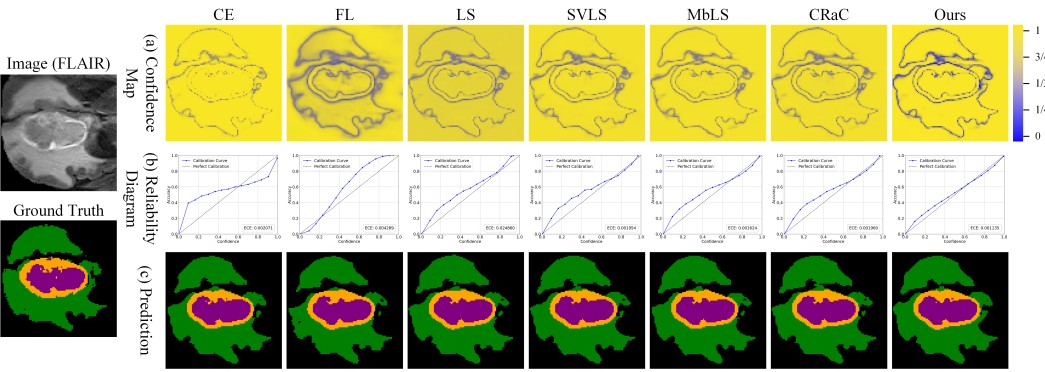

Figure 2: BraTS2020 brain tumor segmentation: (a) confidence maps with blue (low confidence) and orange (high confidence), (b) reliability diagrams showing calibration quality against the ideal diagonal baseline, and (c) segmentation results comparison with black (background), green (edema), purple (non-enhancing necrosis), and orange (enhancing tumor).

**Quantitative Evaluation in Calibration and Segmentation.** Table 1 shows that our method achieves the best and second-best segmentation performance across all metrics, outperforming the CE baseline and leading segmentation methods. As shown in Table 1, our method demonstrates consistently strong and robust performance across Dice, HD95, and ASD metrics, validating its effectiveness across diverse segmentation tasks. As shown in Table 2, our method demonstrates state-of-the-art calibration, achieving the lowest ECE and SCE scores across all datasets. These results highlight the reliability and robustness of our approach in producing well-calibrated confidence estimates across the full probability spectrum. CRaC's superior TACE performance on ACDC and PROMISE2012 stems from its task-specific calibration tailored for fine-grained confidence adjustments in small, well-defined anatomical structures such as heart chambers and the prostate. By leveraging spatial consistency and localized uncertainty modeling, CRaC effectively reduces subtle miscalibrations in high-certainty regions emphasized by TACE. However, this focused calibration strategy can sometimes compromise segmentation accuracy. In contrast, our method achieves a

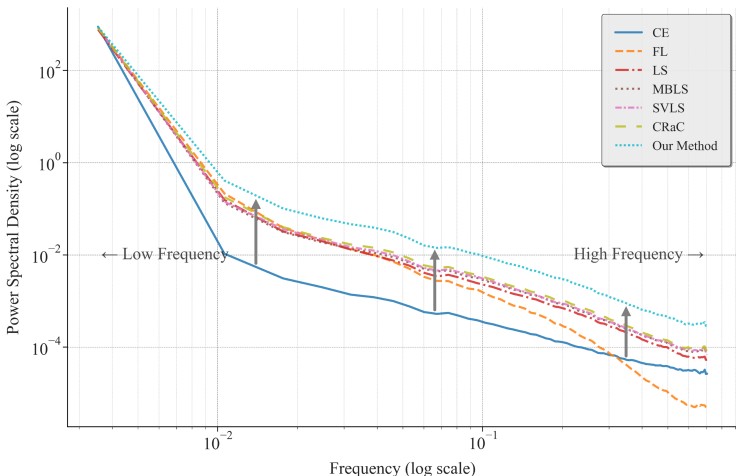

Figure 3: Comparison of power spectral density of confidence maps on the BraTS2020 dataset across different approaches.

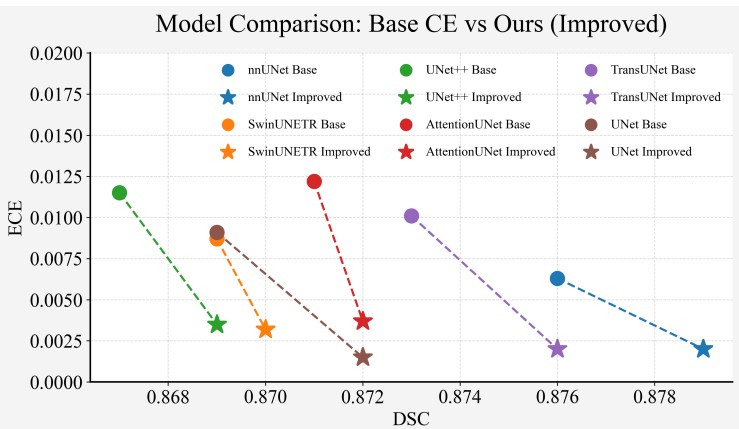

Figure 4: Generalizability of the method across segmentation architectures: improvements in segmentation (DSC) and calibration (ECE) performance compared to the CE baseline.

more balanced calibration reflected in ECE and SCE metrics, alongside consistently superior segmentation performance across diverse datasets.

**Qualitative Analysis in Calibration and Spectral Analysis.** Fig. 8 illustrates how our method enhances uncertainty calibration and segmentation performance on BraTS2020. The CE baseline yields disconnected boundaries in the confidence map (Fig. 8(a)) and shows a mixed calibration behavior in the reliability diagram (Fig. 8(b)): it is underconfident in the low-confidence region, yet becomes systematically overconfident in the high-confidence region (under the ideal curve). FL applies overly aggressive confidence suppression to encourage low-confidence boundaries, which produces diffuse and spatially imprecise uncertainty regions in the confidence map; correspondingly, its reliability curve exhibits consistent underconfidence even in the high-confidence region. Other methods also tend to be underconfident to varying degrees, with underconfidence being more pronounced in the low-confidence region for most approaches. LS follows this general trend and is particularly underconfident at low confidence, while still showing some underconfidence in the high-confidence region. In contrast, our spectral regularization and smoothing promote a balanced frequency representation, enabling sharper and more precise boundary localization in the confidence maps and yielding a reliability curve that stays closest to the ideal diagonal across confidence bins.

Table 3: Ablation study results of our method on the BraTS2020 and FLARE2021 datasets. We evaluate the contribution of each key component across segmentation metrics (DSC, HD95, ASD) and calibration metrics (ECE, SCE, TACE). The configurations are: Baseline ($\mathcal{L}_{CE}$), adding spectral entropy regularization without temporal smoothing ($\mathcal{L}_{CE}$ w/ $\mathcal{L}_{\text{Spectral}}$ w/o $W$), and our full method with smoothing ($\mathcal{L}_{CE}$ w/ $\mathcal{L}_{\text{Spectral}}$).

| Configuration | BraTS2020 | | | | | | FLARE2021 | | | | | |
|---|---|---|---|---|---|---|---|---|---|---|---|---|
| | DSC↑ | HD95↓ | ASD↓ | ECE↓ | SCE↓ | TACE↓ | DSC↑ | HD95↓ | ASD↓ | ECE↓ | SCE↓ | TACE↓ |
| Baseline ($\mathcal{L}_{CE}$) | 0.869 | 2.27 | 0.40 | 0.0091 | 0.0107 | 0.0427 | 0.915 | 8.29 | 2.64 | 0.0255 | 0.0181 | 0.0378 |
| $\mathcal{L}_{CE}$ w/ $\mathcal{L}_{\text{Spectral}}$ w/o $W$ | 0.870 | 2.25 | 0.39 | 0.0065 | 0.0078 | 0.0315 | 0.921 | 6.15 | 1.98 | 0.0170 | 0.0123 | 0.0210 |
| $\mathcal{L}_{CE}$ w/ $\mathcal{L}_{\text{Spectral}}$ | 0.872 | 2.23 | 0.37 | 0.0015 | 0.0008 | 0.0202 | 0.925 | 4.55 | 1.34 | 0.0008 | 0.0006 | 0.0031 |

Fig. 3 provides a spectral perspective on calibration by linking the power spectral density (PSD) distribution to miscalibration patterns observed in the reliability diagram (Fig. 2(b)). The CE baseline exhibits the lowest overall spectral power, reflecting strong confidence saturation and a steep PSD slope that indicates significant spectral imbalance. This spectral bias corresponds to its overconfident predictions in high-confidence regions and underconfident predictions in low-confidence regions in the reliability diagram. Focal Loss reinforces low-frequency dominance, which moderately enhances the PSD in low-frequency regions but sharply suppresses high-frequency components—its high-frequency PSD is even lower than CE. This imbalance hinders its ability to represent boundary-level uncertainty, resulting in underconfident predictions in high-confidence regions. Other methods show comparable overall spectral power and insufficient high-frequency representation, which aligns with their mild underconfidence in the reliability diagram. In contrast, our method achieves the highest overall spectral power and the most balanced frequency distribution, particularly in the high-frequency regions critical for boundary uncertainty. This enables accurate modeling of detailed confidence variations and contributes to the best overall calibration performance on multiple datasets.

To validate the generality of our approach, we applied it across a diverse set of widely adopted segmentation architectures (Fig. 4), including nnUNet Isensee et al. (2021), SwinUNETR Hatamizadeh et al. (2021), UNet++ Zhou et al. (2019), AttentionUNet Oktay et al. (2018), and TransUNet Chen et al. (2021b). Using a fixed set of hyperparameters for all experiments, our method consistently improved both segmentation accuracy and calibration. This result demonstrates the robustness and plug-and-play applicability of our approach.

**Ablation Studies.** Our ablation studies (Table 3) evaluate the contribution of each component. Introducing unsmoothed spectral regularization ($\mathcal{L}_{\text{CE}}$ w/ $\mathcal{L}_{\text{Spectral}}$ w/o $W$) improves both segmentation and calibration metrics over the baseline, confirming the benefit of frequency/domain analysis. However, the gains in boundary/sensitive metrics (HD95, ASD) are modest, suggesting that instantaneous spectral feedback is noisy. Our full method, which adds temporal smoothing of the power spectrum ($\mathcal{L}_{\text{CE}}$ w/ $\mathcal{L}_{\text{Spectral}}$), yields substantial improvements across all metrics. This demonstrates that stabilizing the spectral target is crucial for achieving superior segmentation accuracy and robust calibration.

## 6 CONCLUSION

In this work, we introduced a frequency-aware calibration framework to address the persistent challenge of confidence miscalibration in medical image segmentation. By analyzing model uncertainty through the lens of frequency domain behavior, we identified spectral bias and confidence saturation as two fundamental causes of miscalibration. Our method combines spectral entropy regularization, which promotes a balanced frequency spectrum and increases overall spectral power, with a temporal spectral smoothing module that stabilizes frequency-wise statistics across training batches. This joint design enables the model to more accurately calibrate the predictions of both high-frequency boundary and low-frequency structure. Experiments across six public datasets and multiple segmentation architectures confirm that our approach consistently improves calibration quality without degrading segmentation performance. These results demonstrate the potential of frequency domain calibration as a robust foundation for developing more reliable and trustworthy medical image segmentation systems.

ACKNOWLEDGMENTS

This work was supported by the National Natural Science Foundation of China (No. 62571054).

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

## A   APPENDIX

This appendix provides supplementary materials to support the main manuscript. We offer a more detailed interpretation of the multi-dimensional performance comparison and a comprehensive sensitivity analysis of our method's key hyperparameters. A statement regarding the use of Large Language Models (LLMs) in preparing this manuscript is provided in Section A.1.

### A.1   STATEMENT ON LLM USAGE

In accordance with conference guidelines, we disclose the use of a Large Language Model (LLM) during the preparation of this manuscript. The LLM was employed exclusively for copyediting to improve the clarity, conciseness, and grammatical correctness of the text. All scientific contributions—including research ideation, methodological design, experimental setup, data analysis, and interpretation of results—were conducted entirely by the authors. The LLM did not generate scientific content or insights. The authors have reviewed, edited, and assume full responsibility for the final content of this paper.

### A.2   DATA PREPROCESSING

A standard preprocessing pipeline was applied to all medical image data to ensure consistency across datasets and stable model training. This pipeline was uniformly applied to all methods under comparison to ensure a fair evaluation.

**Image Resampling and Spacing Normalization:** All images were resampled to a unified isotropic voxel spacing, defined as the median voxel spacing of the dataset. Trilinear interpolation was used for intensity images, whereas nearest-neighbor interpolation was applied to segmentation masks to preserve discrete label integrity.

**Intensity Normalization:** For CT images, Hounsfield Unit (HU) values were clipped to a clinically relevant range of [-1000, 400] HU, followed by Z-score normalization (i.e., subtracting the mean and dividing by the standard deviation). For MRI data, Z-score normalization was applied exclusively to foreground voxels (i.e., non-zero values) to prevent background regions from skewing the statistics.

**Data Augmentation:** During training, a suite of data augmentation techniques was employed to enhance model robustness and generalization, including gamma transformation, additive Gaussian noise, Gaussian blurring, and brightness adjustment.

---

**Algorithm 1** Frequency-Aware Calibration Training Procedure

---

**Require:** Model $f_\theta$, training data loader $\mathcal{D}$, loss weight $\lambda$, window size $W$, number of frequency bands $K$.

**Ensure:** Trained model parameters $\theta$.

1: Initialize model parameters $\theta$.
2: Initialize historical spectral power buffer $\mathcal{S}_{\text{hist}} \leftarrow []$.
3: **for** each batch $\{x, y\}$ in $\mathcal{D}$ **do**
4:     **// Forward Pass and Standard Segmentation Loss**
5:     $z \leftarrow f_\theta(x)$                                     ▷ Get logits, shape: $(B, C, D, H, W)$
6:     $\mathcal{L}_{\text{CE}} \leftarrow \text{CrossEntropyLoss}(z, y)$
7:     **// Per-Sample Spectral Analysis**
8:     Initialize batch spectral loss $\mathcal{L}_{\text{Spectral}} \leftarrow 0$.
9:     Let $S_{\text{batch}}$ be an empty list to store spectral vectors for the current batch.
10:     **for** $b = 1, \ldots, B$ **do**                        ▷ Iterate over each sample in the batch
11:         $z_b \leftarrow \max_c z_{b,c}$                   ▷ Generate scalar confidence map from logits
12:         $F_b \leftarrow \text{Shift}(\text{FFT3D}(z_b))$         ▷ Transform to frequency domain
13:         $E_b \leftarrow |F_b|^2$                 ▷ Compute Power Spectral Density (PSD)
14:         Let $S_b = (S_b^{(1)}, \ldots, S_b^{(K)})$ be the spectral power vector for sample $b$.
15:         **for** $k = 1, \ldots, K$ **do**
16:             $S_b^{(k)} \leftarrow \sum_{(u,v,w) \in \mathcal{I}_k} E_b(u, v, w)$     ▷ Aggregate power in frequency band $k$
17:         **end for**
18:         Append $S_b$ to $S_{\text{batch}}$.
19:     **end for**
20:     **// Update Smoothed Spectral Profile and Target Entropy**
21:     $\bar{S}_{\text{batch}} \leftarrow \frac{1}{B} \sum_{b=1}^{B} S_{\text{batch}}[b]$            ▷ Average spectral power for the current batch
22:     Append $\bar{S}_{\text{batch}}$ to $\mathcal{S}_{\text{hist}}$.
23:     **if** $|\mathcal{S}_{\text{hist}}| > W$ **then**
24:         Remove oldest entry from $\mathcal{S}_{\text{hist}}$.
25:     **end if**
26:     $\widetilde{S} \leftarrow \frac{1}{|\mathcal{S}_{\text{hist}}|} \sum_{S' \in \mathcal{S}_{\text{hist}}} S'$         ▷ Compute smoothed spectral profile
27:     $\widetilde{P} \leftarrow \text{Normalize}(\widetilde{S})$             ▷ Convert to probability distribution
28:     $H_{\text{target}} \leftarrow \text{ShannonEntropy}(\widetilde{P})$        ▷ Compute target entropy
29:     **// Compute Spectral Regularization Loss for the Batch**
30:     **for** $b = 1, \ldots, B$ **do**
31:         $S_b \leftarrow S_{\text{batch}}[b]$
32:         $P_b \leftarrow \text{Normalize}(S_b)$
33:         $H_{\text{sample}} \leftarrow \text{ShannonEntropy}(P_b)$
34:         $\text{loss}_b \leftarrow \max(0, H_{\text{target}} - H_{\text{sample}})^2$
35:         $\mathcal{L}_{\text{Spectral}} \leftarrow \mathcal{L}_{\text{Spectral}} + \text{loss}_b$
36:     **end for**
37:     $\mathcal{L}_{\text{Spectral}} \leftarrow \frac{1}{B} \mathcal{L}_{\text{Spectral}}$
38:     **// Update Model Parameters**
39:     $\mathcal{L}_{\text{total}} \leftarrow \mathcal{L}_{\text{CE}} + \lambda \mathcal{L}_{\text{Spectral}}$
40:     Update $\theta$ using gradients from $\mathcal{L}_{\text{total}}$.
41: **end for**

---

### A.3 THEORETICAL MOTIVATION

**From Confidence Maps to 3D Signals.** Our framework formulates the model's confidence map as a 3D signal. Given a 3D medical image, a segmentation model $f_\theta$ produces logits $z(\mathbf{x})$ at each spatial location $\mathbf{x} \in \mathbb{R}^3$. A scalar confidence map, $c(\mathbf{x})$, is derived from these logits (e.g., via the maximum logit value) to represent the model's spatial confidence distribution. The 3D Fourier Transform decomposes this signal into its constituent frequency components:

$$\hat{c}(\mathbf{k}) = \mathcal{F}\{c(\mathbf{x})\} = \iiint_{\mathbb{R}^3} c(\mathbf{x}) e^{-i 2\pi \mathbf{k} \cdot \mathbf{x}} d\mathbf{x} \tag{11}$$

where $\mathbf{k}$ is the 3D frequency vector.

**Spectral Signatures of Segmentation Quality.** The Power Spectral Density (PSD), $E(\mathbf{k}) = |\hat{c}(\mathbf{k})|^2$, quantifies the distribution of the signal's energy across frequencies. By Parseval's theorem, the total energy is conserved between the spatial and frequency domains. Low-frequency components correspond to large, smooth structures in $c(\mathbf{x})$, whereas high-frequency components capture sharp transitions and fine details. An effective model should produce confidence maps that are smooth within homogeneous regions (low frequencies) but sharp at anatomical boundaries (high frequencies). A common failure mode, particularly for over-confident models, is the suppression of high-frequency detail, yielding overly smooth or "blob-like" predictions. This manifests as a PSD pathologically concentrated in the low-frequency bands.

**Spectral Entropy as an Information-Theoretic Regularizer.** We employ spectral entropy to quantify the structural complexity of the confidence map. The frequency space is partitioned into $K$ disjoint bands $\{\mathcal{I}_k\}_{k=1}^K$, and the power in each band is defined as $S^{(k)} = \int_{\mathbf{k} \in \mathcal{I}_k} E(\mathbf{k}) d\mathbf{k}$. Normalizing these values yields a probability distribution $P = \{P^{(k)}\}_{k=1}^K$, where $P^{(k)} = S^{(k)} / \sum_j S^{(j)}$. The spectral entropy is the Shannon entropy of this distribution:

$$H(P) = -\sum_{k=1}^K P^{(k)} \log(P^{(k)} + \varepsilon) \tag{12}$$

This metric aligns with the **Principle of Maximum Entropy**, where a higher-entropy distribution is more uniform and embodies fewer structural assumptions. Low spectral entropy indicates a degenerate, low-information output (e.g., over-smoothed) characterized by an overly simplistic confidence structure.

**Objective as a Corrective Regularizer.** Our objective does not maximize entropy directly but instead acts as a corrective measure against spectral collapse. The loss, $\mathcal{L}_{\text{Spectral}} \propto \max(0, H_{\text{target}} - H_{\text{sample}})^2$, penalizes a sample only if its spectral entropy $H_{\text{sample}}$ falls below a dynamic target $H_{\text{target}}$. This design is theoretically motivated to enforce a minimum level of structural complexity, thereby preventing the aforementioned failure mode without discouraging valid, spectrally rich predictions. It steers the model away from degenerate solutions characterized by overconfidence and excessive structural simplicity.

## A.4 Algorithm Pseudocode

Algorithm 1 details the proposed training procedure. Each training batch undergoes three main steps:

1. **Per-Sample Spectral Analysis:** Following a standard forward pass and cross-entropy loss computation, a scalar confidence map is generated from the maximum logits for each sample. This map is transformed into the frequency domain, and its PSD is aggregated into $K$ frequency bands to produce a spectral power vector.

2. **Dynamic Target Generation:** Spectral power vectors from the current batch are averaged and appended to a historical buffer of size $W$. Averaging this buffer yields a temporally smoothed spectral profile, providing a stable estimate of the desired spectral characteristics. The Shannon entropy of this profile defines the dynamic target entropy for the current iteration.

3. **Regularization and Optimization:** The spectral entropy of each sample is compared against the dynamic target. A hinge-like loss penalizes samples with entropy below this target, thereby promoting a more informative spectral distribution. The final objective is a weighted sum of the cross-entropy loss and this spectral regularization loss.

## A.5 Comprehensive Performance Comparison

Fig. 5 presents a holistic performance evaluation, visualizing the trade-off between segmentation accuracy and uncertainty calibration. In this visualization, superior methods enclose a larger, more regular polygon. The ideal method would maximize this area, indicating superior performance

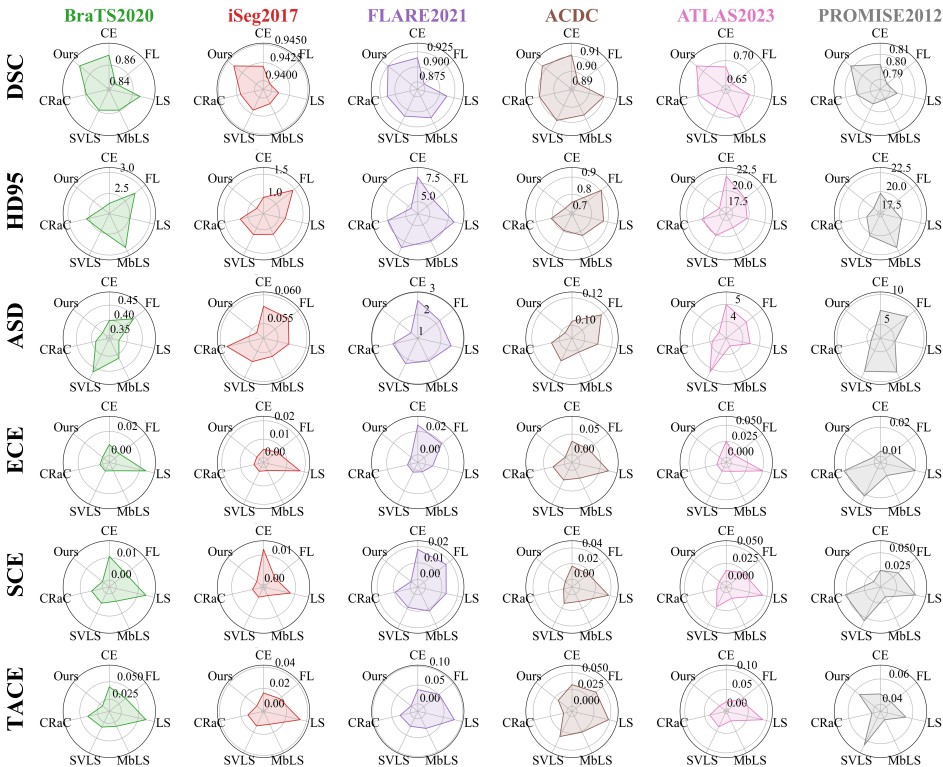

Figure 5: Multi-dimensional radar chart comparing eight methods across segmentation (DSC, HD95, ASD) and calibration (ECE, SCE, TACE) metrics. A larger, more regular area indicates better overall performance.

across all metrics. Our proposed method exhibits a notably balanced and expansive shape, demonstrating consistently high segmentation scores (high DSC, low HD95/ASD) and strong calibration performance (low ECE/SCE/TACE). In contrast, competing methods often display "spiky" profiles; for instance, some achieve a high DSC at the cost of significantly worse calibration metrics (e.g., ECE, TACE), indicating poor uncertainty estimation. This visual analysis underscores the primary strength of our approach: achieving a robust balance between segmentation accuracy and uncertainty calibration, a critical requirement for clinical applications where both prediction correctness and confidence reliability are paramount.

## A.6 HYPERPARAMETER SENSITIVITY ANALYSIS

Table 4 presents a sensitivity analysis of the key hyperparameters, revealing consistent trends across the BraTS2020 and FLARE2021 datasets. The regularization weight $\lambda$ exhibits a clear trade-off: small values provide insufficient regularization (high ECE), whereas excessively large values degrade segmentation performance (low DSC). Optimal values (e.g., $\lambda = 0.01$ for BraTS2020, $\lambda = 0.05$ for FLARE2021) balance both objectives effectively. Similarly, the temporal smoothing window $W$ is crucial for a stable entropy target. A small window ($W = 10$) creates a noisy target, whereas a large one ($W = 100$) adapts slowly to distributional shifts; a moderate size ($W \in [25,75]$) achieves the best results. Finally, the number of frequency bands $K$ determines the granularity of spectral analysis. Too few bands ($K = 2$) oversimplify the spectrum, while too many ($K = 10$) are sensitive to noise. A value between 3 and 7 provides a robust balance, consistent with related work Bo et al. (2025). These results confirm that the parameter response is stable and predictable, providing clear guidance for application to new datasets.

Table 4: Sensitivity analysis of the key hyperparameters $(\lambda, W, K)$ on the BraTS2020 and FLARE2021 datasets.

| Varying Hyperparameter | Value | BraTS2020 | | FLARE2021 | |
|---|---|---|---|---|---|
| | | DSC↑ | ECE↓ | DSC↑ | ECE↓ |
| $\lambda$ (Fixed $W = 50, K = 5$) | 0.001 | 0.870 | 0.0045 | 0.922 | 0.0028 |
| | 0.005 | 0.871 | 0.0020 | 0.924 | 0.0015 |
| | 0.010 | 0.872 | 0.0015 | 0.925 | 0.0008 |
| | 0.050 | 0.871 | 0.0013 | 0.925 | 0.0007 |
| | 0.100 | 0.865 | 0.0012 | 0.920 | 0.0008 |
| $W$ (Fixed $\lambda = 0.01, K = 5$) | 10 | 0.870 | 0.0025 | 0.922 | 0.0014 |
| | 25 | 0.871 | 0.0018 | 0.924 | 0.0010 |
| | 50 | 0.872 | 0.0015 | 0.925 | 0.0008 |
| | 75 | 0.872 | 0.0016 | 0.925 | 0.0009 |
| | 100 | 0.871 | 0.0022 | 0.923 | 0.0013 |
| $K$ (Fixed $\lambda = 0.01, W = 50$) | 2 | 0.870 | 0.0028 | 0.923 | 0.0016 |
| | 3 | 0.871 | 0.0019 | 0.925 | 0.0008 |
| | 5 | 0.872 | 0.0015 | 0.925 | 0.0009 |
| | 7 | 0.871 | 0.0017 | 0.924 | 0.0011 |
| | 10 | 0.870 | 0.0023 | 0.922 | 0.0015 |

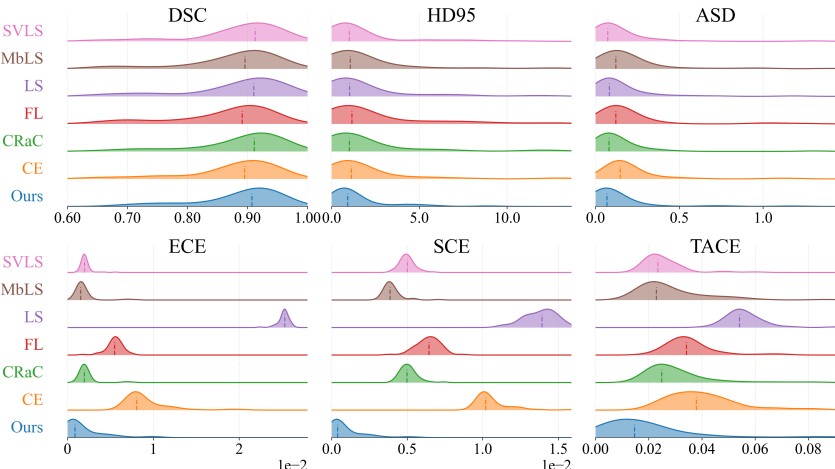

Figure 6: Distribution analysis of segmentation (DSC) and calibration (ECE) metrics on the BraTS2020 dataset. Our method demonstrates improved mean performance and reduced variance compared to the baseline.

## A.7 DISTRIBUTION ANALYSIS

Fig. 6 illustrates the performance distributions for the Dice Score (DSC) and Expected Calibration Error (ECE). Our method's distributions exhibit two key characteristics: (1) a rightward shift in the mean for DSC and a leftward shift for ECE, indicating superior average performance, and (2) reduced variance (a narrower, taller peak). The latter is particularly important, as it demonstrates improved model robustness and reliability. The smaller spread signifies a reduction in outlier predictions and catastrophic failures, where a baseline model might perform poorly on challenging cases. This enhanced consistency is a direct result of our spectral regularization, which prevents the model from adopting overly simplistic, spectrally biased solutions and ensures more reliable performance across the test population.

## A.8 COMPREHENSIVE PERFORMANCE RANKING

This analysis evaluates the generalization capabilities of our method across multiple datasets and metrics. As shown in Fig. 7, we aggregate performance ranks to provide a holistic comparison. For aggregation, ranks are assigned to each method on every dataset-metric pair (rank 1 is best) and then

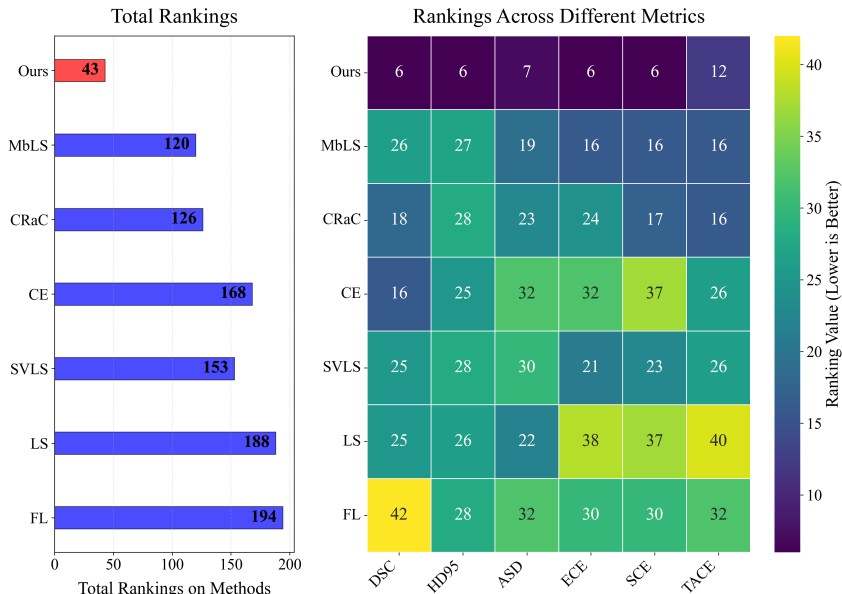

Figure 7: Cross-dataset performance ranking. Left: Aggregate ranking scores (lower is better). Right: Heatmap of individual metric ranks (darker is better). Our method achieves a consistently high rank across diverse metrics and datasets.

summed to yield an overall score (lower is better). The left panel confirms our method's superior aggregate rank. The heatmap on the right is more revealing: competing methods often exhibit specialized performance, excelling on certain metrics or datasets while faltering on others, suggesting potential overfitting to specific data characteristics. In contrast, our method achieves consistently strong performance across the board. This demonstrates that by operating in the frequency domain, our regularizer targets a fundamental aspect of the model's confidence representation rather than dataset-specific spatial statistics, leading to a more robust and holistically effective model.

## A.9 VISUALIZING THE MITIGATION OF SPECTRAL BIAS

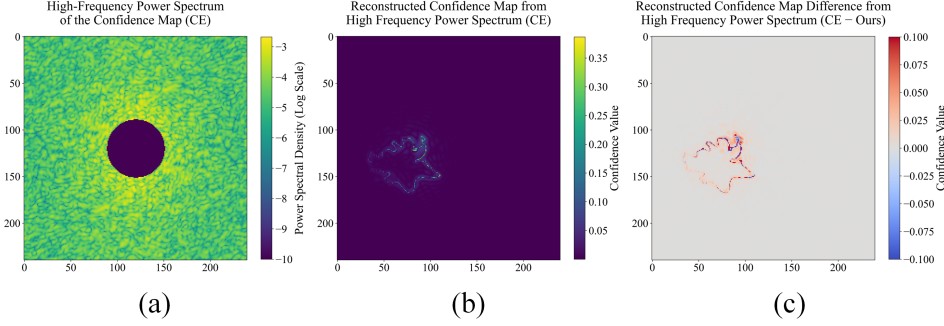

Figure 8: Analysis of high-frequency components on a BraTS2020 sample. (a) High-frequency power from the baseline model's logits. (b) Confidence map reconstructed from (a), showing misplaced high-confidence "hotspots." (c) Confidence difference map (Baseline − Ours), where red indicates overconfidence corrected by our approach, correlating strongly with the hotspots in (b).

This analysis provides a direct visual link between the theoretical motivation of spectral bias and our method's corrective effect. We isolate the high-frequency components from a baseline model's logit output for a challenging sample. Applying an inverse Fourier transform to these components reveals their spatial manifestation: misplaced, high-confidence hotspots, demonstrating how

unchecked high-frequency power can lead to erroneous predictions at complex boundaries (Fig. 8a-b). The confidence difference map (Baseline − Ours) in Fig. 8c highlights regions where our method reduces overconfidence. The strong spatial correlation between these regions and the baseline's hotspots serves as direct evidence that our spectral regularization is working as intended. It successfully suppresses problematic high-frequency power, leading to better-calibrated and more structurally coherent confidence maps, particularly in anatomically complex regions.

