# OpenReview forum: "Rethinking Model Calibration through Spectral Entropy Regularization in Medical Image Segmentation"
_ICLR.cc/2026/Conference — ICLR 2026 Poster_

### Official Review · Reviewer_LwpE · 2025-10-28

**Soundness:** 3
**Presentation:** 3
**Contribution:** 3
**Rating:** 6
**Confidence:** 4

**Summary:**

This paper addresses the critical problem of overconfidence in deep neural networks for medical image segmentation. The authors propose a novel approach that reframes model calibration from a frequency-domain perspective. They identify two primary causes of miscalibration: spectral bias, where models over-prioritize low-frequency components, and confidence saturation, where overconfidence leads to a general suppression of power spectral density (PSD).

To solve this, the paper introduces a frequency-aware calibration framework consisting of two main components:
1. Spectral Entropy Regularization: A novel loss term that encourages a more balanced and informative frequency spectrum in the model's confidence maps (derived from logits).
2. Power Spectral Smoothing: A temporal smoothing module that stabilizes the spectral statistics used to create a dynamic target for the regularization loss.

The authors conduct an extensive evaluation on six public medical imaging datasets and multiple segmentation architectures. Their results show that the proposed method significantly improves calibration metrics (ECE, SCE) while simultaneously maintaining or even improving segmentation accuracy (DSC, HD95).

**Strengths:**

1. The paper's primary strength is its highly original approach. Reframing the calibration problem through the lens of spectral bias and confidence saturation is a novel and insightful contribution. While frequency-domain methods exist, this appears to be the first work to build a coherent framework specifically for confidence calibration by regularizing the output's intrinsic spectral properties. The results are highly significant. The method achieves state-of-the-art calibration, showing massive and consistent improvements in ECE and SCE across all six datasets (Table 2).

2. The technical quality of the submission is exceptionally high. The link between overconfidence and a suppressed, low-frequency-dominated PSD is clearly motivated and visualized (Fig. 1) . The methodology is also well-justified, including the sound rationale for operating on logits rather than saturated softmax probabilities. This technical rigor extends to the evaluation, which is both comprehensive and robust. The authors validate their method on six diverse datasets, compare it against five strong baselines, and use a full suite of segmentation and calibration metrics.

3. Furthermore, the method's "plug-and-play" nature is convincingly demonstrated, showing consistent improvements across six different segmentation architectures, including U-Net, nnUNet, and SwinUNETR (Fig. 4) . The analysis is also thorough; the ablation study (Table 3) clearly isolates the contribution of each component, and the appendix provides further strong evidence (e.g., radar charts, rank heatmaps). Crucially, this work breaks the common accuracy-calibration trade-off, often improving segmentation performance while fixing calibration (Table 1)

**Weaknesses:**

1. The weaknesses are minor and do not undermine the contribution.

2. The paper does not quantify the computational overhead of the proposed method.

3. Calculating 3D FFTs, aggregating power, and managing a historical buffer for each batch introduces a non-trivial training cost compared to the baseline, and this information would be valuable for other researchers. Additionally, the entire spectral analysis relies on a scalar confidence map derived from the maximum logit value.

4. This simplification discards information from other classes, and it remains unclear how this approach would scale to problems with many classes.

5. While the sensitivity analysis for $K$ (the number of frequency bands) is good, the paper could benefit from a more qualitative discussion of why $K=5$ provides an optimal balance, perhaps by linking the bands to specific anatomical structures.

**Questions:**

1. Could you please provide an analysis of the training-time overhead? For example, what is the percentage increase in training time per epoch when adding the proposed regularization framework compared to the standard cross-entropy baseline?

2. The temporal smoothing uses a simple sliding window of size $W$. Did you consider using an Exponential Moving Average (EMA) instead? An EMA might offer a more flexible trade-off between stability and adaptability, especially in non-stationary settings where the "target" spectral profile might evolve.

3. The method relies on the max-logit to create the confidence map. Have you experimented with other scalar representations of confidence, such as the predictive entropy of the softmax output? It would be interesting to see if its spectral properties provide a similar or complementary signal for regularization.

4. The motivation in Figure 1 defines a "well-calibrated" boundary pixel as having $\alpha=0.5$ (maximum uncertainty). This is intuitive for a binary foreground/background decision. How does this intuition translate to a multi-class boundary, for instance, the boundary between "edema" and "enhancing tumor" in the BraTS dataset (which has 4 classes)? Is the goal still maximum uncertainty, or is the ideal spectral signature more complex in this scenario?

---

> ### Author Response · Authors · 2025-11-27
> **Response to Reviewer LwpE**
>
> We sincerely thank the reviewer for the highly positive assessment.
> 1. Analysis of Training-Time Overhead
>
> We clarify that the computational overhead introduced by our framework is negligible, typically amounting to 3-5% of the total training time per epoch. The backbone segmentation networks (e.g., SwinUNETR) involve computationally intensive operations like self-attention during both forward and backward passes. In contrast, our spectral branch operates only on the final output confidence map (a single-channel tensor of shape B×1×D×H×W). The 3D Fast Fourier Transform (FFT) has an O(NlogN) complexity, making it extremely efficient on modern GPUs (e.g., cuFFT). Consequently, the added cost is marginal compared to the heavy computational load of the primary segmentation task.
>
> 2. Temporal Smoothing
>
> We thank the reviewer for suggesting EMA as an alternative and agree that it is a natural choice for temporal smoothing. We chose a sliding window of size $W$ to better handle the strongly non-stationary regime at the beginning of training, where spectral statistics change rapidly. EMA has "infinite memory," meaning early, poorly calibrated spectral profiles could theoretically influence the target for a long time, introducing lag. A sliding window of size W (e.g., 50 iterations) provides a "hard cutoff," ensuring the target entropy is determined solely by the most recent, relevant training dynamics, allowing the target to adapt quickly without being dragged down by outdated historical data.
>
> 3. Choice of Scalar Representation
>
> We deliberately use the max-logit confidence map rather than predictive entropy or softmax probabilities as the basis for our spectral analysis. The maximum logit is directly tied to overconfidence, and using raw logits avoids the dynamic-range compression introduced by the softmax nonlinearity. Once probabilities saturate toward 0 or 1, their spectra become dominated by low-frequency plateaus and lose the subtle high-frequency variations around boundaries that we aim to regularize. By contrast, the max-logit map preserves these fine-grained variations, providing a richer signal for frequency-aware calibration. Note that this choice affects only the regularization branch: the full multi-class logit tensor is still used to train the segmentation head in the usual way, so the method scales to many classes even though the calibration operates on a scalar confidence proxy. We agree that entropy-based or multi-scalar variants could provide complementary information, and we plan to explore such combinations in future work
>
> 4. Interpretation of "Maximum Uncertainty" in Multi-Class Scenarios
>
> We agree that multi-class boundaries can, in principle, induce more complex spectral signatures. In practice, however, the local neighborhood of a boundary voxel is usually dominated by two competing classes (e.g., edema vs. enhancing tumor in BraTS), so the binary intuition largely carries over. A well-calibrated model should predict similar probabilities for those two classes (e.g., $\mathrm{P(Edema)} \approx \mathrm{P(ET)} \approx 0.5$), yielding a max probability close to 0.5 at that interface. Our goal is not to enforce maximum uncertainty globally, but to allow lower confidence and higher spectral entropy specifically at genuinely ambiguous interfaces, while keeping high confidence in clearly separable regions. From a spectral viewpoint, an overconfident, step-like transition creates a low-entropy spectrum dominated by low frequencies, whereas a smoother confidence transition at the boundary yields richer high-frequency content. Our regularizer encourages this latter behavior, mitigating flat overconfident plateaus and leading to better-calibrated multi-class boundaries.
>
> 5. Qualitative Rationale for K=5 Frequency Bands
>
> Beyond the quantitative sensitivity analysis, we chose $K=5$ because it provides a useful decomposition that loosely aligns with anatomical scales. The lowest band captures global organ shape; the middle bands focus on sub-structures and regional variations; the highest bands capture fine textures and sharp boundary changes. With very small $K$ (e.g., 2), global shape and boundary details are entangled in the same bands, making it harder for the regularizer to selectively target boundary uncertainty. With very large $K$ (e.g., 10), individual bands become too narrow and sensitive to image-specific noise. Empirically, $K=5$ offered the best trade-off in Table 4: it enabled us to emphasize high-frequency boundary bands without disrupting the learning of stable low-frequency organ shape.

---

### Official Review · Reviewer_dMRR · 2025-11-01

**Soundness:** 3
**Presentation:** 2
**Contribution:** 3
**Rating:** 4
**Confidence:** 4

**Summary:**

This paper introduces a power spectral smoothing mechanism for calibrating segmentation models. Specifically, it proposes a spectral entropy regularisation to improves frequency balance in confidence maps, involving spectral decomposition, window smoothing, leading to better model calibration results.

**Strengths:**

Handle the calibration task from the spectral domain is novel;
The paper is easy to follow.

**Weaknesses:**

1) Motivation is not solid;
2) Too many sophisticated hyper-parameters to tune;
3) Missing important baselines (Temperature scaling, Platt scaling) and metrics (IoU).

**Questions:**

1) In Fig.1, why a lower confidence score at boundary indicates better calibrated outputs? Especially if this is for ground truth labels.
2) Calibration is important for non-segmentation tasks, but how is this important for segmentation? One will have to set a threshold for the segmentation results to be useful, so the accuracy/IoU naturally becomes the focus.
3) Is there any reason that in experiments Temperature scaling/Platt scaling and IoU is not report? Temperature scaling/Platt scaling may require validation set but since there are already so many hyper-parameters have been tuned this should be doable.

---

> ### Author Response · Authors · 2025-11-27
> **Response to Reviewer dMRR**
>
> We thank the reviewer for recognizing the novelty of our work and for finding the paper easy to follow.
>
> 1. Clarification on Motivation and Figure 1
>
> We respectfully clarify that our motivation stems from a critical observation: spatial-domain losses often fail due to the "spectral bias" of neural networks, where models preferentially learn low-frequency structures and neglect high-frequency details, leading to overconfidence at boundaries.
> As detailed in Section 3.1 and Figure 1, spatial calibration methods (like label smoothing) can dampen confidence globally but fail to distinguish between structural certainty and boundary ambiguity.
> In medical datasets, the “ground truth” is usually provided as a single binary mask, but it often aggregates multiple experts (e.g., majority vote, STAPLE fusion). Especially near lesion or organ boundaries, annotators frequently disagree due to partial-volume effects, low contrast, and annotation experience. As a result, substantial multi-expert disagreement is collapsed into deterministic 0/1 labels; from a probabilistic viewpoint these boundary labels are inherently noisy and are better treated as noisy supervision that should be handled more cautiously during training.
> In Figure 1: at a fuzzy boundary where the true chance of being foreground is around 50%, a calibrated model should output a probability close to 0.5 rather than 0 or 1, so lower confidence at boundaries actually reflects better calibration even though the stored ground-truth mask is binary.
> Our spectral analysis reveals that "confidence saturation" manifests as a collapse of high-frequency power in the confidence map. Therefore, the motivation is not merely to apply frequency operations, but to explicitly regularize the model to maintain the spectral complexity required to represent uncertainty correctly. By enhancing the high-frequency components in the confidence maps, our frequency-aware regularization directly targets boundary overconfidence and makes the link between boundary miscalibration and spectral properties concrete and quantitatively measurable in the frequency domain.
>
> 2. Hyperparameters
>
> While our method involves three hyperparameters (λ, W, K), we demonstrate in Appendix A.6 (Sensitivity Analysis) and Table 4 that the method is highly robust and does not require delicate tuning. The window size (W) acts as a stabilizer for statistics, and results show consistently stable performance for W ∈ [25, 75]. The number of frequency bands (K) follows standard signal processing practices for partitioning spectrums, and performance is consistent for K values between 3 and 7. Importantly, we used a fixed set of hyperparameters across all six diverse datasets (from brain MRI to cardiac CT) and six different network architectures (Figure 4), yet consistently achieved state-of-the-art results. This empirical evidence confirms that the parameters are generalizable and do not require per-dataset fine-tuning to be effective.
>
> 3. Baselines (In-training vs. Post-hoc) and Metrics
>
> We focused our comparison on in-training calibration methods (Focal Loss, Label Smoothing, CRaC, etc.) because our goal is to learn better feature representations that are inherently calibrated, rather than patching the output of a fixed model. Post-hoc methods like Temperature Scaling (TS) and Platt Scaling, while useful, have significant limitations in medical imaging: they are dataset-level scalars that cannot handle the spatial heterogeneity of uncertainty within an image, and they often fail under domain shifts. Regarding metrics, we reported the DSC and HD95 because these are the standard, primary metrics for the medical image segmentation community. Since DSC and IoU are monotonically related (DSC = 2IoU/(1+IoU)), improvements in DSC mathematically guarantee improvements in IoU.
>
> 4. Importance of Calibration in Segmentation
>
> We agree that a threshold (e.g., 0.5) must ultimately be applied and that overlap metrics such as Dice/IoU remain the primary indicators of segmentation quality, but our goal is to ensure that the probabilities being thresholded are themselves reliable (e.g., a predicted 0.9 corresponds to ≈90% correctness). In clinical oncology, segmentation models are typically used with a human in the loop: oncologists inspect uncertainty maps to identify regions where the model is unsure, verify tumor boundaries for treatment. Well-calibrated confidence maps are therefore essential, because they ensure that truly ambiguous regions are highlighted rather than hidden behind overconfident predictions. Thus, while Accuracy/DSC measures final overlap, calibration metrics such as ECE measure the reliability of the model’s risk estimates, which is critical for patient safety. At tumor boundaries in particular, voxel-wise class decisions are crucial for effective treatment and reduced side effects, making reliable uncertainty in these regions essential.

---

### Official Review · Reviewer_2Bao · 2025-11-01

**Soundness:** 2
**Presentation:** 2
**Contribution:** 3
**Rating:** 6
**Confidence:** 4

**Summary:**

This paper investigates the problem of confidence miscalibration in medical image segmentation and provides a new perspective by analyzing it in the frequency domain. The authors identify two core causes—spectral bias (models’ preference for low-frequency components) and confidence saturation (loss of spectral power in confidence maps)—that contribute to overconfident predictions. To tackle these issues, the paper introduces a frequency-aware calibration framework combining spectral entropy regularization and power spectral smoothing. Experiments across six public datasets and multiple segmentation architectures demonstrate that the proposed method consistently improves calibration metrics while maintaining segmentation accuracy.

**Strengths:**

Novel Perspective: The work provides an original frequency-domain interpretation of model miscalibration, which is rarely explored in medical image analysis.

Theoretical Insight: The identification of spectral bias and confidence saturation as underlying causes is conceptually sound and empirically validated.

Methodological Innovation: The proposed spectral entropy regularization and power spectral smoothing are well-motivated, simple to implement, and complementary.

Comprehensive Evaluation: Experiments on multiple datasets and architectures show consistent improvements in calibration without accuracy degradation.

Clinical Relevance: By enhancing uncertainty reliability, the approach contributes directly to improving trustworthiness in medical AI systems.

**Weaknesses:**

While the proposed frequency-aware calibration framework is conceptually interesting, several aspects require further clarification and discussion. The frequency-domain operations in this work are quite similar to those in [1][2], which also explore the distinction between low- and high-frequency components; however, the paper does not explicitly analyze or compare with these prior studies. Moreover, key training details are missing—particularly the batch size used and how it affects frequency statistics. Performing repeated frequency-domain transformations across multiple batches could introduce considerable computational overhead. The penalty minimization is said to promote a more balanced spectral distribution by transferring energy from dominant low-frequency to underrepresented high-frequency regions, but the paper does not explain how this spectral balance is quantitatively measured, nor whether it might degrade segmentation accuracy. Finally, in class-imbalanced scenarios (e.g., when certain anatomical categories have far more samples), it remains unclear whether the proposed approach retains its effectiveness.
[1]Feng, Wei, et al. "Unsupervised domain adaptation for medical image segmentation by selective entropy constraints and adaptive semantic alignment." Proceedings of the AAAI Conference on Artificial Intelligence. Vol. 37. No. 1. 2023.
[2]Chen, Guangyao, et al. "Amplitude-phase recombination: Rethinking robustness of convolutional neural networks in frequency domain." Proceedings of the IEEE/CVF international conference on computer vision. 2021.

**Questions:**

N/A

---

> ### Author Response · Authors · 2025-11-27
> **Response to Reviewer 2Bao**
>
> We sincerely thank the reviewer for the constructive feedback and for recognizing the novelty of our frequency-domain perspective.
>
> 1. Distinction from Prior Frequency-Domain Works
>
> We appreciate the reviewer pointing out these related works. Although [1] and [2] also exploit low- vs. high-frequency structure, their goals and operating points differ fundamentally from ours. Feng et al. [1] employ entropy constraints for Unsupervised Domain Adaptation (UDA) to align feature distributions across domains, and Chen et al. [2] utilize Amplitude-Phase Recombination (APR) on input images/features to disentangle style from content for robustness. In contrast, our method operates directly on the output confidence maps (logits) in a fully supervised setting, with the explicit objective of improving uncertainty calibration rather than domain alignment or robustness. We do not manipulate input images or intermediate feature maps to separate style; instead, we analyze the spectral density of the prediction itself to mitigate "confidence saturation." As detailed in Section 3.1 and Figure 1, our core insight is that overconfident predictions manifest as a loss of high-frequency power in the confidence map. Therefore, our spectral entropy regularization is designed to restore this lost spectral information and thereby improve calibration, a mechanism distinct from the alignment or robustness goals of the cited works. We are willing to add these two paper into our related work in Section .
>
> 2. Training Details, Batch Size, and Frequency Statistics
>
> Regarding the training configuration, we use a batch size of 2 for all experiments (Section 4.1, “Implementation Details”), which is standard under 3D medical segmentation memory constraints. We agree that such small batches can lead to noisy instantaneous frequency statistics and this is precisely why we introduce Power Spectral Smoothing (Section 3.2). Instead of relying on a single batch, we maintain a historical window (e.g., $W=50$, validated in Table 4) to aggregate spectral power over recent iterations and form a temporally smoothed spectral profile. This smoothed profile then serves as the target for our spectral entropy regularization, decoupling the calibration objective from per-batch noise and ensuring stable frequency statistics even when $N=2$.
>
> 3. Computational Overhead of Frequency-Domain Transformations
>
> We clarify that the computational cost introduced by our method is negligible relative to the backbone segmentation network. The Fast Fourier Transform (FFT) is applied only to the final scalar confidence map (shape B × 1 × D × H × W), not to high-dimensional feature volumes, so the additional work is an $O(N\log N)$ operation on a single channel. In our implementation, this adds approximately 3–5% training-time overhead per epoch compared with the cross-entropy baseline, while inference is unchanged because no spectral operations are used at test time. Given that backbones such as SwinUNETR dominate both FLOPs and memory, we view this extra cost as modest relative to the observed calibration gains.
>
> 4. Measurement of Spectral Balance and Impact on Segmentation Accuracy
>
> The "spectral balance" is quantitatively measured using Spectral Entropy (H_spec), defined in Eq. 8, computed over the band-wise power spectrum defined in Eq. 4. Higher $H_{\text{spec}}$ indicates a more uniform distribution of power across frequency bands and explicitly counters the low-frequency–dominated “spectral bias” observed in Figure 1. Empirically, Tables 1 and 2 show that enforcing this spectral balance improves calibration metrics (ECE, SCE) while maintaining or slightly improving segmentation accuracy (DSC, HD95) across all six datasets, rather than degrading it. This is consistent with our intuition in Section 3.3: encouraging non-degenerate high-frequency content in the confidence maps forces the network to model sharper yet better calibrated boundaries instead of producing over-smoothed, overconfident blobs.
>
> 5. Effectiveness in Class-Imbalanced Scenarios
>
> Class imbalance is indeed pervasive in medical segmentation, and several of our benchmarks are strongly imbalanced. For example, in BraTS2020 the three tumor subregions (WT, TC, ET) occupy only a small fraction of the brain volume, and in FLARE2021 many organs occupy a tiny fraction compared with background. Our regularizer is computed on the confidence map derived from the maximum logit (Eq. 1), so the spectral entropy is driven by the spatial pattern of high-confidence predictions rather than by class frequency itself. The improvements we observe on these datasets—especially reduced ECE/SCE alongside stable or better DSC for small tumor regions—suggest that the method remains effective in class-imbalanced settings and does not rely on balanced class distributions.

---

### Meta-Review · Area_Chair_dbF9 · 2025-12-17

**Summary:**

Reviewers found the frequency-domain perspective on calibration novel and clinically relevant, with strong empirical results across multiple datasets and architectures demonstrating consistent improvements in calibration without sacrificing segmentation accuracy.

However, several reviewers raised concerns about the clarity of motivation, missing comparisons to standard post-hoc calibration baselines, and insufficient discussion of computational overhead and hyperparameter complexity.

Metareview thinks Reviewer dMRR's negative comments are minor, and have been sufficiently addressed by authors. Therefore, the metareview recommends acceptance (borderline accept) of this paper.

**Reviewer Concerns:**

The rebuttal addressed several key concerns by clearly distinguishing the proposed method from prior frequency-domain works, clarifying the role of batch size and introducing power spectral smoothing to stabilize frequency statistics, quantifying the computational overhead, and explaining how spectral entropy is measured and why it does not harm segmentation accuracy. It also justified the relevance of calibration for segmentation and clarified robustness under class imbalance.

Remaining concerns include the absence of explicit comparisons to standard post-hoc calibration baselines, the reliance on several hyperparameters despite demonstrated robustness, and limited discussion of how the approach generalizes to many-class settings beyond the max-logit confidence proxy.

**Reviewer Scores:**

Reviewer 2Bao already rated the paper marginally above the acceptance threshold (6) and, given that their main concerns (distinction from prior frequency-domain work, batch size effects, overhead, class imbalance) were directly addressed in the rebuttal, would likely maintain or slightly increase their score.

Reviewer dMRR expressed concerns about motivation and missing post-hoc baselines, while many conceptual questions were clarified, the lack of temperature/Platt scaling comparisons likely keeps their score unchanged (4).

Reviewer LwpE was positive and had mainly minor concerns about overhead and design choices, all of which were answered quantitatively, so their score would likely remain high or increase slightly (6).

---

### Decision · Program_Chairs · 2026-01-26

Accept (Poster)